# An impulse-based derivation of the Kutta-Joukowsky equation for wind turbine thrust

Eric J. Limacher[1] and David H. Wood[2]

[1]Department of Mechanical Engineering, Federal University of Pará, Belém, Brazil
[2]Department of Mechanical and Manufacturing Engineering, University of Calgary, Calgary T2N 1N4, AB, Canada.

**Correspondence:** David H. Wood (dhwood@ucalgary.ca)

**Abstract.** Using the concept of impulse in control volume (CV) analysis, we derive an equation for steady wind turbine thrust in a constant, spatially uniform wind, dependent on the circumferential velocity and tip speed ratio. Viscous drag on the blades is neglected. Although the resulting equation has been known for a long time, the present derivation offers novel insights. A major advantage of an impulse formulation is that it removes the pressure and introduces the vorticity, allowing unambiguous use of the equation immediately behind the blades. We assume that the vorticity is frozen relative to an observer rotating with the blades, so the vortex lines follow the local streamlines in the rotating frame. The impulse analysis shows that radial velocity should approach the value of the axial induction in the vicinity of the rotor tip. Nonetheless, by assuming both the radial and axial velocities are continuous through the rotor disk, their contributions to the final thrust expression cancel. The final integral equation for thrust can be viewed as a generalization of the Kutta-Joukowsky theorem for the rotor forces, and we prove that the conditions under which the Kutta-Joukowsky equations are exactly valid for individual blade elements are high tip speed ratio and/or high number of blades. The present analysis also suggests that when either of these conditions hold, one additional condition is sufficient for blade-element independence of the conventional thrust equation: constancy of the vortex pitch across the wake. The exact derivation of the Kutta-Joukowsky equation also shows that the pressure acting on the expanding flow upwind of the rotor does not contribute to blade element thrust. Finally, we derive the exact relationship between power and thrust and show that the common approximation that rotor power is the product of the velocity through the rotor and the thrust becomes true only at high tip speed ratio.

## 1 Introduction

Blade element theory (BET) for wind turbines uses the fundamental assumption that the forces acting on the elements comprising the rotor blades are given by the Kutta-Joukowsky theorem. The thrust and torque are balanced by the change in the axial and angular momentum, respectively, of the flow through a control volume (CV) enclosing the rotor; the combination of BET and momentum theory gives rise to blade-element momentum (BEM) theory. BEM is developed in all texts on wind turbine aerodynamics, such as Burton *et al.* (2011), Wood (2011), and Hansen (2015), so it is unnecessary to repeat it here. Three assumptions of BEM will be examined herein through the lenses of impulse theory and helical vortex theory: (i) the

radial velocity is negligible at the rotor, or at least has little effect on blade forces, (ii) the Kutta-Joukowsky equation is applicable to individual blade elements, and (iii) pressure on the lateral control surfaces does not contribute to blade-element thrust (commonly referred to as the assumption of blade-element independence).

Impulse theory is a body of work in which a component of fluid-dynamic force is expressed in terms of the first moment of vorticity, e.g. Wu et al. (2015):

$$\frac{1}{N-1}\int_V \mathbf{x} \times \mathbf{\Omega} dV, \tag{1}$$

where $V$ is some fluid volume, $N$ is the dimension of the space, $\mathbf{x}$ is the position vector, and $\mathbf{\Omega}$ is the vorticity vector. The use of the word "impulse" may confuse the unfamiliar reader; it is an imprecise yet well-established nomenclature. Lighthill (1986) showed that the first moment of vorticity is equal to the impulse (i.e. the integral of force applied over time) necessary to establish an unbounded vortical flow from rest (the domain is unbounded, but the region of vortical fluid remains bounded). Since that time, the term "impulse" has been co-opted to refer specifically to the first moment of vorticity, as recounted in section 2.5 of Limacher (2019), and we will continue to use this terminology here.

The introduction of the concept of impulse removes the pressure and introduces vorticity to the equations of linear momentum conservation. This allows, for example, the impulse equation to be used in the immediate vicinity of the blades. The thrust equation that results from this analysis, which depends only the azimuthal velocity behind the rotor, is equivalent to expressions found in Glauert (1935) and Sørensen (2016), but the derivation to be presented here yields novel insights regarding the radial velocity in the vicinity the rotor. The derived equation is then manipulated to address points (ii) and (iii) above.

Impulse methods have been used to remove the pressure from force balances in fluid mechanics since their introduction by Thomson (1882) to determine the speed of a vortex ring. In their review, Wu et al. (2015) recount that exact impulse-based expressions for aerodynamic force were derived independently by Burgers (1921), Wu (1981) and Lighthill (1986). Discussions of impulse formulations can also be found in Lamb (1932), Batchelor (1967), and Saffman (1992). They have recently gained popularity for use with planar or volume meaurements of fluid velocities (but not pressure) from particle image velocimetry and related techniques, e.g. Rival & van Oudheusden (2017) and Limacher et al. (2018). For example, Limacher et al. (2019a) tested an impulse equation for thrust against measurements of an impulsively started circular cylinder using PIV data, and this method was then compared by Limacher et al. (2020) to a momentum-based formulation. This study uses the formulation of Noca et al. (1997) which is developed in detail by Noca (1997); we recommend the latter to any reader interested in details. Derivation of the impulse formulation begins with the conventional application of the Reynolds transport theorem to the momentum equation, the application of the so-called "impulse-momentum identity" to replace momentum with impulse (Equation (3.1) of Noca (1997)), and proceeds by removing the pressure using the Euler or Navier-Stokes equations.

The present work will consider a wind turbine rotating steadily at a tip speed ratio, $\lambda$, defined as the ratio of the circumferential velocity of the blade tips to the wind speed. The latter velocity will be used to normalize all velocities. All lengths are normalized by rotor radius, $R$. An inertial frame of reference will be used in the body of the paper, but an alternative derivation

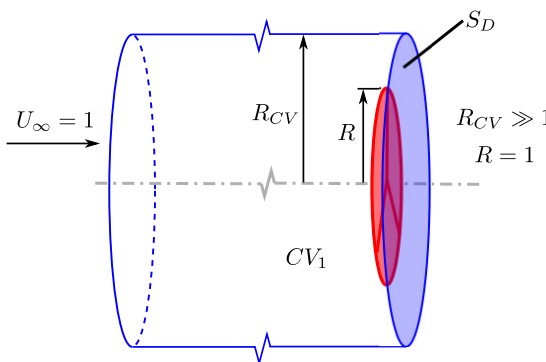
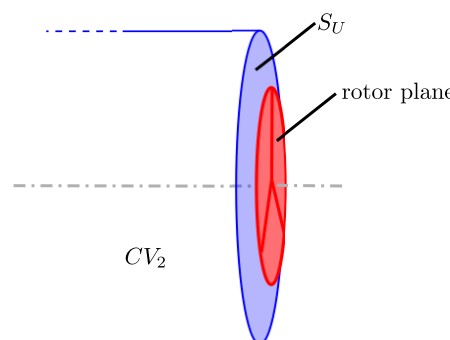

downstream CS just behind rotor

downstream CS just ahead of rotor

**Figure 1.** Control volumes (CVs) to be used in the present analysis. In both variants, the upstream face is well upstream of the rotor, where the velocity is equal to the freestream, and $R_{CV} \gg R$. The downstream control surface is just downstream and just upstream of the rotor plane in $CV_1$ and $CV_2$, respectively, and the corresponding donwstream control surfaces (CS) are labelled $S_D$ and $S_U$.

in a frame of reference that rotates with the blades is provided in Appendix A. We start by using two variants of a cylindrical CV, both of whose upwind faces are well ahead of the blades, and whose radius $R_{CV} \gg R$. This ensures that the streamwise velocity along the surface at $R_{CV}$ is equal to the wind speed. The downstream control surface is either just downstream or just upstream of the rotor plane; these variants are called $CV_1$ and $CV_2$, respectively, as shown in figure 1.

For this analysis, we assume that the forces acting on the blades and blade elements are generated entirely by the radial vorticity "bound" to the blades and we ignore any radial vorticity in the wake resulting from viscous drag on the blades. The distinction is illuminated by considering a stationary rotor whose blades are formed from a symmetric airfoil. If the blade chord aligns with the flow at all radii, no bound vorticity is generated and no torque acts on the rotor. The equation we derive below for thrust would not hold in this case as the thrust would be due entirely to viscous drag manifested as radial vorticity in the wake. Further,

if the circumferential extent of the radial vorticity in the wake is significant, then the axial velocity will not be continuous across the rotor. The impulse analysis can be extended to this case but that would make a complex introductory description even more complex. In stationary cylindrical polar co-ordinates $(x, \theta, z)$, where $x$ is the radius, $\theta$ is in the circumferential direction, and $z$ the axial co-ordinate, the velocities sufficiently far upwind of the rotor are $(0, 0, 1)$. Immediately ahead of the rotor they are $(v, 0, 1 - a)$, where $a$ is the standard axial induction factor, and $v$ is the radial velocity through the rotor. Immediately behind

the rotor, the velocities are $(v, 2w, 1 - a)$, where $w$ is the velocity induced at the blades by the wake. We use the term "wake" to describe the part of the flow that has passed through the rotor and we will assume that $w$ and all components of vorticity are zero outside the wake.

Impulse equations are derived from conventional CV analysis based on the Reynolds transport theorem. For an inertial CV in an incompressible fluid, the equation for force, $\mathbf{F}$, is given in all fluid mechanics texts and by Noca's (1997) Equation (3.36):

$$\frac{\mathbf{F}}{\rho} = -\frac{d}{dt}\int_V \mathbf{U}dV + \oint_S \mathbf{n}\cdot\left(-\frac{P}{\rho}\mathbf{I} - \mathbf{U}\mathbf{U} + \mathbf{T}\right)dS \tag{2}$$

where $t$ is time, and $\mathbf{U}$ is the velocity vector, $S$ is the control surface bounding the control volume, $V$, $\mathbf{n}$ is the outward-facing normal on $S$, $\mathbf{I}$ is the unit tensor, $P$ is the pressure and $\mathbf{T}$ is the viscous stress tensor. The $\mathbf{n}\cdot\mathbf{U}\mathbf{U}$ term gives the conventional momentum deficit (MD) when the equation is used to determine thrust.. By applying the impulse-momentum identity, and removing $P$ using the Navier-Stokes equations, Noca obtained his Equations (3.55) and (3.56), which we combine as

$$\frac{\mathbf{F}}{\rho} = -\frac{1}{N-1}\frac{d}{dt}\int_V \mathbf{x}\times\boldsymbol{\Omega}dV + \oint_S \mathbf{n}\cdot\left(\frac{1}{2}U^2\mathbf{I} - \mathbf{U}\mathbf{U} - \frac{1}{N-1}\mathbf{U}(\mathbf{x}\times\boldsymbol{\Omega}) + \frac{1}{N-1}\boldsymbol{\Omega}(\mathbf{x}\times\mathbf{U})\right)dS + \mathbf{F}_v \tag{3}$$

where $N = 3$ for our analysis, $\frac{1}{2}U^2 = \frac{1}{2}\mathbf{U}\cdot\mathbf{U}$ is the kinetic energy per unit mass, and $\mathbf{F}_v$ is shorthand for the viscous terms evaluated on $S$. In common with other CV analyses of wind turbines, $\mathbf{F}_v$ will be ignored. Note that the conventional MD term appears in both (2) and (3). Of major importance for wind turbine aerodynamics is that the removal of $P$ introduces the vorticity. The first term on the right is the time derivative of the impulse, whose contribution to thrust will vanish for the steady flows analyzed here. We will, however, continue to describe our analysis as an "impulse" one because it follows the same general path, and because, though not immediately obvious, the remaining terms are related to the impulse in the domain outside of $V$, Wu et al. (2015), Kang et al. (2017). The two vortex terms (the two surface integrals involving vorticity) may be non-zero in both steady and unsteady flow.

The next section applies Equation (3) to yield simplified expressions of wind turbine thrust, and section 3 briefly discusses angular momentum and angular impulse. The reader interested primarily in the consequences of the present impulse analysis for BEM may skip ahead to section 4. Finally, our conclusions are presented in section 5.

## 2 The Impulse Equation for Thrust in Stationary Co-ordinates

The cylindrical CV of radius $R_{CV}$ in figure 1 begins sufficiently far from the rotor that no influence of the rotor occurs on the inlet face. The downwind exit face of $CV_1$, which is used exclusively in this section, is just behind the rotor, as shown on the left-hand side of figure 1. The time derivative of the impulse integral in (3) does not necessarily vanish identically, but its thrust contribution does; since the vortical wake rotates with the blades, only the direction (but not the magnitude) of the impulse integral can change, and its time derivative reduces to

$$\frac{d}{dt}\int_{CV_1} \mathbf{x}\times\boldsymbol{\Omega}dV = \boldsymbol{\Lambda}\times\int_{CV_1} \mathbf{x}\times\boldsymbol{\Omega}dV. \tag{4}$$

$\boldsymbol{\Lambda}$ is the rotation vector; for steady rotation $\boldsymbol{\Lambda} = \lambda\mathbf{e_z}$ where $\mathbf{e_z}$ is the unit vector in the $z-$direction. Therefore, the thrust contribution vanishes since $\boldsymbol{\Lambda}$ is parallel to the thrust direction. The MD term is the conventional one, derived in any textbook

on wind turbine aerodynamics:

$$\int_S u(1-u)dS = \int_{S_D} a(1-a)dS \tag{5}$$

where $S_D$ denotes the downwind face of the CV. The axial velocity contributes to the second (kinetic energy) term an amount

$$-\frac{1}{2}\int_{S_D}(1-u^2)dS. \tag{6}$$

Equations (5) and (6) can be combined to give

$$-\frac{1}{2}\int_{S_D}a^2 dS. \tag{7}$$

The kinetic energy has a contribution from the radial velocity,

$$\frac{1}{2}\int_{S_D}v^2 dS, \tag{8}$$

and from the circumferential velocity,

$$2\int_{S_D}w^2 dS \tag{9}$$

The radial velocity exiting the horizontal face of the CV at $R_{CV}$ does not contribute to the kinetic energy because $v(R_{CV}) \sim 1/R_{CV}$ so the integral of $v^2$ over the cylindrical face can be made arbitrarily small by making $R_{CV}$ sufficiently large. Vorticity can only be non-zero on the downstream face of the CV and the vortex terms simplify to

$$\frac{1}{2}\int_{S_D}(1-a)x\Omega_\theta dS - \frac{1}{2}\int_{S_D}2\Omega_z wx dS. \tag{10}$$

After substitution of Equations (7) through (10) into (3), the thrust becomes

$$\frac{T}{\rho} = \int_{S_D}v^2 dS - \int_{S_D}a^2 dS + 4\int_{S_D}w^2 dS + \frac{1}{2}\int_{S_D}(1-a)x\Omega_\theta dS - \frac{1}{2}\int_{S_D}2\Omega_z wx dS, \tag{11}$$

where all velocities and vorticities are expressed relative to an inertial frame. The assumption of a rigid wake, with vortex lines coinciding with streamlines in the rotating frame of reference, yields the relationships

$$\frac{2w+\lambda x}{1-a} = \frac{\Omega_\theta}{\Omega_z} = \frac{x}{p}, \tag{12}$$

where $p$ is the vortex pitch. When diffusion occurs, this condition need not hold over the whole wake. However, assuming that fluid elements retain constant circulation once they are shed from the rotor (i.e. Kelvin's theorem), Equation (12) is necessary to prevent distortion of the wake structure. Using Equation (12), the vortex terms can be rewritten as

$$\frac{1}{2}\int_{S_D}\left[(1-a)\Omega_\theta - (2w+\lambda x)\Omega_z\right]xdS + \frac{1}{2}\int_{S_D}\Omega_z\lambda x^2 dS. \tag{13}$$

The terms in the square brackets of the first integrand cancel by Equation (12). The vortex terms thus reduce to the remaining term in (13), which can be interpreted as a thrust due to wake rotation. We now proceed to manipulate this term to express it in terms of velocity. Substituting $dS = x\,dx\,d\theta$ into the previous expression, the term becomes

$$\frac{1}{2}\lambda \int_0^{2\pi}\int_0^{\infty} \Omega_z x^3 \, dx \, d\theta. \tag{14}$$

The streamwise vorticity in polar coordinates is

$$\Omega_z = -2\left(\frac{dw}{dx} + \frac{w}{x}\right) - \frac{1}{x}\frac{\partial v}{\partial \theta}, \tag{15}$$

where the negative sign on the $w$ terms is due to our chosen sign convection, whereby positive $w$ corresponds to positive torque. When substituted into (14), the contribution of $\partial v/\partial\theta$ vanishes:

$$-\frac{1}{2}\lambda \int_0^{\infty} x^2 \left[\int_0^{2\pi} \frac{\partial v}{\partial \theta} d\theta\right] dx = 0. \tag{16}$$

The vortex terms thus become

$$-\lambda \int_0^{2\pi} \left[\int_0^1 \frac{dw}{dx} x^3 \, dx + \int_0^1 w x^2 \, dx\right] d\theta. \tag{17}$$

After integrating the first term by parts, and assuming $wx^3 \to 0$ as $x \to \infty$, the whole expression becomes

$$2\lambda \int_0^{2\pi}\int_0^{\infty} w x^2 \, dx \, d\theta = 2\lambda \int_{S_D} w x \, dS. \tag{18}$$

Replacing the vortex terms in Equation (11) with this expression, we arrive at

$$\frac{T}{\rho} = \frac{1}{2}\int_{S_D} (v^2 - a^2)\,dS + 2\int_{S_D} (w^2 + \lambda w x)\,dS. \tag{19}$$

Since a future goal of this research program is unsteady turbine modelling, casting the present analysis in the rotating frame, as described in Appendix A, may prove useful.

## 3 Angular Momentum and Angular Impulse

The other dynamic conservation equation applied in conventional CV analysis of wind turbines is for $\boldsymbol{\tau} = \tau\mathbf{e_z}$, the torque on the rotor. In general vector form, this is expressed as

$$\frac{\boldsymbol{\tau}}{\rho} = -\frac{d}{dt}\int_{V_m} \mathbf{x}\times\mathbf{U}\,dV \tag{20}$$

$$= -\frac{d}{dt}\int_{V} \mathbf{x}\times\mathbf{U}\,dV - \oint_{S} \mathbf{n}\cdot\mathbf{U}(\mathbf{x}\times\mathbf{U})\,dS \tag{21}$$

where $V_m$ is a material volume. Assuming the velocity field in the CV to be stationary relative to the blades, the axial component of the first integral vanishes, as previously shown for the impulse derivative term using Equation (4). Equation (21) for the rotor torque then gives the conventional result:

$$\frac{\tau}{\rho} = 2 \int_{S_D} (1-a)xw dS. \tag{22}$$

Angular momentum conservation can also be expressed in terms of angular impulse, but, as we will briefly demonstrate, this formulation is unnecessarily complicated; since pressure does not contribute to the angular momentum balance on a cylindrical CV, the advantage lent by the impulse approach to thrust does not apply to torque.

The identity relating angular momentum to angular impulse, given in Equation (3.3.9) of Wu et al. (2015), can be differentiated with respect to time to yield

$$\frac{d}{dt} \int_{V_m} \mathbf{x} \times \mathbf{U} dV = -\frac{1}{2}\frac{d}{dt} \int_{V_m} |\mathbf{x}|^2 \mathbf{\Omega} dV + \frac{1}{2}\frac{d}{dt} \oint_{S_m} |\mathbf{x}|^2 \mathbf{n} \times \mathbf{U} dS, \tag{23}$$

where $S_m$ is the boundary of the material volume, and the first term on the right-hand side is the time derivative of angular impulse. It is tempting to assume that the last integral will vanish, and that the integral on the right-hand side of Equation (20) can be replaced by the derivative of angular impulse, expressing the torque in terms of the trailing vorticity crossing $S_D$. Substitution of the Navier-Stokes equation for the material derivative of $\mathbf{U}$ in the last integral of (23) will yield an integrand

of the form $|\mathbf{x}|^2 \mathbf{n} \times \mathbf{\nabla}P$, whose integral will vanish on the closed contour of an axisymmetric CV, and viscous effects can be neglected as before. Deformation of the material surface, however, gives rise to derivatives of $|\mathbf{x}|^2$ and $\mathbf{n}$ that do not vanish in general. In the special case that axial velocity is constant on the downstream face $S_D$, the second integral does vanish and Equation (22) can be recovered from the angular impulse term after integration by parts. However, the assumption of constant axial velocity on $S_D$ was unnecessary in the standard angular momentum approach, and we thus find that the concept of angular

impulse has not enriched the analysis.

## 4   Impulse and Blade Element Momentum Theory

### 4.1   Radial velocity and blade element independence

If the downwind face of the CV is moved to be just ahead of the rotor, to give the first use of $CV_2$ defined in figure 1, $T$ in Equation (19) becomes zero (since no body is enclosed by the CV), as does the last term on the right-hand side since the

azimuthal velocity is zero ahead of the rotor (Taylor (1921)). Thus,

$$0 = \int_{S_U} \left( v^2 - a^2 \right) dS. \tag{24}$$

For any actuator disk, Equation (24) implies that the magnitude of $v$ and $a$ must be equal over at least some of the wake. More specifically, it can be shown that the integral over the wake is negative and positive over the external flow, where $v > a$. As a

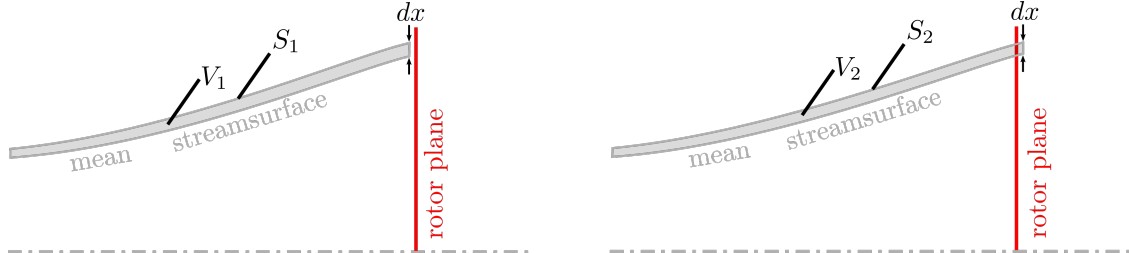

**Figure 2.** Annular CVs used for the blade-element force analysis. Lateral surfaces align with mean streamsurfaces. The downstream face of width $dx$ is just upstream and just downstream of the rotor for $V_1$ and $V_2$, respectively.

likely consequence, $v \approx a$ at the edge of the wake, as shown by Figures 15 and 16 of Madsen et al. (2010) and Figure 3.2 of Sørensen (2016). This implies a significant radial deflection of the streamlines in the tip region as they pass through the rotor, but we are unaware of any study of the effect of this "crossflow" on blade element forces. Simple expressions for crossflow alterations to airfoil lift and drag are developed by Hodara & Smith (2014). For a turbine with a finite number of blades, the situation is more complex. The early near-wake measurements of Ebert & Wood (2001) and the recent ones of Eriksen & Krogstad (2017), when converted to co-ordinates rotating with the blades, show large positive and negative values of $v(\theta)$ particularly in the tip region. The circumferential average $v$ or the value at the blades, can, therefore, be small while the average of $v^2$ can be significant. The flow in the tip region of a rotor with a finite number of blades is complex, and can include motion of the tip vortex towards, rather than away from the axis of rotation, van Kuik et al. (2014).

Assuming that $a$ and $v$ are continuous through the rotor, Equation (24) applies when the CV is changed to $CV_2$, so that

$$\frac{T}{\rho} = 2 \int_{S_D} \left( w^2 + \lambda w x \right) dS. \tag{25}$$

To determine the blade element version of (25), we consider two variants of an annular CV with lateral surfaces coinciding with mean streamsurfaces: one with the downstream face just upwind of the rotor ($V_1$), and one with it just downwind ($V_2$), as shown in figue 2. In the former case, the flow everywhere in the control volume and on the bounding control surface ($S_1$) is irrotational, and since there is no body enclosed by this CV, we obtain

$$\oint_{S_1} \mathbf{n} \cdot \left( \frac{1}{2} U^2 \mathbf{I} - \mathbf{U} \mathbf{U} \right) dS = 0. \tag{26}$$

Assuming the radial and axial velocities are continuous across the rotor, their contribution to the same integrand over $S_2$ also vanishes, yielding a thrust contribution dependent on azimuthal velocity alone:

$$\oint_{S_2} \mathbf{n} \cdot \left( \frac{1}{2} U^2 \mathbf{I} - \mathbf{U} \mathbf{U} \right) dS = 2 \int_{S'_D} w^2 dS, \tag{27}$$

where $S'_D$ is the downstream face of the annular CV, $V_2$. Since the trailing vortices pierce $S'_D$, the vortex terms contribute $2\lambda w x$ to the integrand of the thrust integral, as in Equation (19). The vortex terms will have a contribution from the lateral

control surfaces, since they will also be pierced by vorticity in the vicinity of the blades. In these terms, one may find a way to express the aerodynamic effect of radial velocity; however, since the present analysis has treated the rotor as a disk, and blade geometry has not been considered, this deeper analysis is left for future work. Neglecting these effects, the expression for the thrust on a blade element becomes

$$\frac{1}{\rho}\frac{dT}{dx} = 2\int\limits_{0}^{2\pi} \left(w^2 + \lambda w x\right) x\, d\theta, \tag{28}$$

which is equivalent to expressions given by Glauert (1935), Sørensen (2016) and van Kuik (2018), derived by them by means of the unsteady Bernoulli equation. Our own version of that derivation has been provided in Appendix B for ease of reference, and it requires the following assumptions: there is no radial vorticity in the wake, which rotates rigidly with the rotor, and the axial and radial velocities are continuous across the disk. We have arrived at Equation (28) using these same assumptions. By reverting to Equation (11), however, we also have an expression that permits a wake of finite thickness, but have not yet explored this extra complication. In Appendix C, we provide an alternative derivation of Equation (28) which gives further information on the role of the pressure on the expanding annular streamtubes.

The impulse form of the blade element thrust equation has nothing equivalent to a pressure term, and thus does not suffer from the omission of pressure forces on the expanding streamtubes in the conventionally derived thrust equation which is given in terms of $u$ rather than $w$, Goorjian (1972). That is, Equation (28) is valid for any blade element regardless of the loading on neighbouring elements. This does not imply the general independence of blade elements in the conventional thrust equation, but the conditions under which Equation (28) reduces to that equation are sufficient conditions for blade-element independence as commonly understood. These conditions are not the commonly-assumed ones.

The most important condition is constancy of vortex pitch across the wake, since any variation in $p$ means that the axial velocity at blade element position $x_{BE}$ is determined partly by the vortex structure for $x > x_{BE}$, and not solely by the local momentum change . The reason is that the average $u$ within a helical vortex is proportional to $1/p$, Kawada, (1936), Hardin (1982), and thus blade element independence requires $p$ to be constant across the wake. If the flow is assumed circumferentially uniform (such as when the number of blades tends to infinity), Equation (12) applies everywhere on $S_D$ and Equation (28) can be rewritten as:

$$\frac{1}{\rho}\frac{dT}{dx} = 2\int\limits_{0}^{2\pi} \left[(1-a)\frac{wx}{p} - w^2\right] x\, d\theta. \tag{29}$$

If $p$ is constant across the wake, and the trailing wake is treated as a collection of non-expanding helicoidal surfaces, e.g. Okulov and Sørensen (2008), it follows from the analysis of a single helical vortex by Kawada (1936) and Hardin (1982) that

$$\frac{x}{p} = \frac{a}{w}. \tag{30}$$

Equation (30) is a necessary condition for the non-deformation of the translating helicoidal surfaces in the far wake, where the circumferential induced velocity is assumed to be half that in the near wake (since the induced velocity from the bound circulation must be of equal magnitude to yield zero circumferential velocity upstream of the rotor). After substitution of Equation (30) into Equation (29), and after conventional normalization, we arrive at

$$\frac{dC_T}{dx} = 8[a(1-a) - w^2]x. \tag{31}$$

As $\lambda \to \infty$, $w^2 \to 0$ and the familar expression from axial momentum theory is recovered;

$$\frac{dC_T}{dx} = 8a(1-a)x, \tag{32}$$

which is Equation (3.41) in Burton et al. (2011) and (3.2) in Wood (2011).

The present analysis suggests that conventional blade element independence is most likely to be valid for a high number of blades, high tip speed ratio, and constant vortex pitch across the wake. Fortunately, a constant-$p$ wake defines an optimal rotor, (originally from Betz (1919), as cited in Okulov and Sørensen (2008)), and thus the typical assumption of blade-element independence is most appropriate at the target operating point. However, this does not necessarily hold for sub-optimal operating conditions, for which performance evaluation remains practically necessary. One solution is to replace Equation (32) with Equation (28) in blade-element analysis. When coupled with Equation (22) for angular momentum, we have two equations for two unknowns – $a$ and $w$ – and the problem is soluble. This approach circumvents the limitations of the conventional blade-element independence assumptions, as Equation (28) requires only that the near wake is thin and rotates rigidly with the blades, which will plausibly hold across operating conditions at typical Reynolds numbers.

### 4.2 Kutta-Joukowsky equation for blade-element forces

We now consider the conditions under which the Kutta-Joukowsky thrust equation is valid for blade elements. Equation (31) assumes that viscous traction has little effect on the trailing vorticity (and thus on the blade forces), and with the additional assumption of azimuthal uniformity, Equation (31) becomes

$$\frac{1}{\rho}\frac{dT}{dx} = (w + \lambda x)\int_0^{2\pi} 2wx\,d\theta. \tag{33}$$

The sum of blade circulations, $\sum_i \Gamma_i$, at a given radial station is equal to the integral on the right-hand side, as can be derived by Stokes' theorem as depicted in figure 3. Taking $S_a$ to be a cylindrical surface of radius $x$ intersecting the blades, with $S_a$ bound externally by $C_a$,

$$\sum_i \Gamma_i = \int_{S_a} \Omega_r\,dS = \oint_{C_a} \mathbf{U} \cdot d\mathbf{l} = \int_0^{2\pi} 2wx\,d\theta, \tag{34}$$

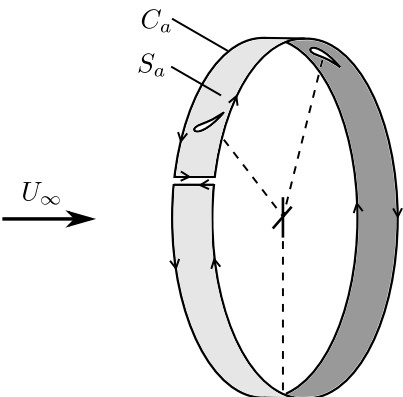

**Figure 3.** The sum of blade circulations at radial station $x$ can be calculated as an area integral of radial vorticity over $S_a$, or a closed line integral over $C_a$, where only the downstream circular section of $C_a$ lends a non-zero contribution.

where $\Omega_r$ is the radial vorticity on $S_a$, $d\mathbf{l}$ is the tangential unit vector along $C_a$, and only the downstream portion of $C_a$ lends a non-zero contribution the contour integral (since azimuthal velocity is zero upstream of the blades, e.g. Taylor (1921)). When all blades are evenly loaded, Equations (33) and (34) combine to yield

$$\frac{1}{\rho}\frac{dT}{dx} = N\Gamma_{BE}\left(w + \lambda x\right) \tag{35}$$

where $N$ is the number of blades and $\Gamma_{BE}$ is the bound circulation of the blade element. Azimuthal uniformity is approached as $N \to \infty$, but Equation (35) is approximately true for finite $N$ when $\lambda x \gg w$, as circumferential non-uniformity in $w$ becomes relatively small. This occurs for the power producing region of the blades at sufficiently high $\lambda$. Equation (22) for the torque can similarly be rewritten in terms of $\Gamma_{BE}$ when $a$ is assumed to be azimuthally uniform:

$$\frac{1}{\rho}\frac{d\tau}{dx} = N\Gamma_{BE}(1-a)x. \tag{36}$$

Equations (35) and (36) are the usual form of the Kutta-Joukowsky theorem for the forces on a blade element. In blade element analysis, the Kutta-Joukowsky equation is usually introduced as an assumption; the present analysis shows this assumption to be valid as $N \to \infty$ and/or $\lambda \to \infty$. Aspects of the nonlinearity in the BEM equations at low $\lambda$ are analyzed by Wood and Okulov (2017).

### 4.3    Relationship between power and thrust

Equation (29) also allows a re-examination of the relationships between $T, \tau$, and turbine power, $P = \lambda\tau$, at least for large $N$. Using (35) and (36) along with (12), we obtain

$$(1-a)\frac{dT}{dx} = \frac{1}{2}\left(\lambda + \frac{1-a}{p}\right)\frac{d\tau}{dx}. \tag{37}$$

Thus for stationary rotors, $dT/dx = 1/(2p)d\tau/dx$. We are completing a study of stationary rotors which includes use of this equation and will report the results separately. One consequence of (37), however, is worth mentioning here: the pitch of the

 vortex system of a stationary rotor must be finite and the vortex lines cannot be straight as is assumed in the lifting line theory of wings. As $\lambda \to \infty$, Equation (12) shows that $(1-a)/p \to \lambda$ at least when $x$ is large in the power-producing region of the rotor, so that $dP/dx \approx \Omega d\tau/dx \approx (1-a)dT/dx$. The relationship $P = (1-a)T$, or its blade element form $dP/dx = (1-a)dT/dx$, is often assumed to hold at all operating conditions, but this analysis shows that it can be accurate only at high $\lambda$. A further consequence of this result is that the analysis developed in this paper is incapable of describing the runaway state of a wind turbine, which requires $P \downarrow 0$ but $T$ to remain finite, usually at high $\lambda$. This limitation likely arises from the breakdown of the steady helical vortex structure directly behind the blades at high $\lambda$, as viewed in the rotating frame of reference, and this hypothesis deserves further investigation.

## 5   Summary and Conclusions

This paper describes the application of the impulse formulation of Noca (1997) to determine the torque and thrust on a steadily rotating wind turbine in a steady, spatially uniform wind. The impulse analysis starts with the usual Reynolds transport theorem applied to the momentum equation for a finite CV and proceeds by removing the pressure. This has two major attractions for wind turbine analysis: (a) the resulting equation can be applied easily anywhere in the flow and (b) the role of the trailing vorticity becomes clear. We assume that the vorticity field is steady when viewed by an observer rotating with the blades, so the vortex lines and streamlines are aligned in this frame. This alignment simplifies the contribution of the trailing vorticity to the thrust expression, and the resulting term can be interpreted as a contribution due to the rotation of the vortical wake. By assuming the axial velocity to be continuous across the rotor, we also assume that the blade forces are generated entirely by the vorticity bound to the rotor, so that, for example, the viscous drag of the elements is ignored. We consider the thrust equation both for the whole rotor and for the individual blade elements. The equation for the latter shows that the radial velocity must have the same magnitude as the axial induction factor over at least some of the wake and exceeds the factor in the external flow.

The equation for rotor thrust and its differential form for a blade element (Equations (25) and (28), respectively), involving the azimuthal but not the axial velocity, have been known for a long time. Of advantage in the present derivation are the more general expressions prior to assuming a thin wake, an assumption which cannot be avoided in existing derivations based on the unsteady Bernoulli equation. Equation (28) can be applied to individual blade elements regardless of the loading on neighbouring elements. This does not necessarily imply the blade-element independence of the conventional thrust equation in terms of the axial velocity, but the present analysis suggests that the following conditions are sufficient: negligible viscous drag on the rotor, constancy of the wake vortex pitch, high number of blades and high tip speed ratio. It was also proven that when the number of blades and/or the tip speed ratio is sufficiently large, the Kutta-Joukowsky equation gives the blade element forces, as is assumed in standard blade element analysis.

Finally, the new thrust equation was used to examine the relationship between power, torque, and thrust. We showed that assuming power to be the product of the axial velocity and thrust can only be approximately correct at high tip speed ratio,

with the consequence that the present analysis cannot describe the runaway state of wind turbines at high tip speed ratio. The analysis also requires the trailing vorticity for stationary rotors to have finite pitch. Thus the trailing vorticity of stationary rotors must be helicoidal as opposed to the straight trailing vortices assumed in lifting line theory for wings.

## Appendix A: The Impulse Equation for Thrust in Steady Rotating Co-ordinates

When the polar co-ordinates are attached to the blades rotating at $\lambda$, the velocities corresponding to those used in the main text are $(0, \lambda x, 1)$ well upwind, $(v, \lambda x, 1-a)$ immediately upwind of the rotor, and $(v, u'_\theta, 1-a) = (v, 2w + \lambda x, 1-a)$ downwind. Equation (3) requires modification to be expressed in terms of velocities and vorticities evaluated in the rotating frame of reference, $\mathbf{U}'$ and $\mathbf{\Omega}'$, respectively. The momentum integral in Equation (2) is replaced by

$$\frac{d}{dt} \int_V \mathbf{U} dV = \frac{d}{dt} \int_V \mathbf{U}' dV + \int_V 2\mathbf{\Lambda} \times \mathbf{U}' dV + \int_V \mathbf{\Lambda} \times (\mathbf{\Lambda} \times \mathbf{x}) dV. \tag{A1}$$

In Noca's derivation of Equation (3) from (2), the pressure, $P$, was removed by using the Navier-Stokes equations in the form of his Equation (3.44):

$$\frac{\partial \mathbf{U}}{\partial t} = -\nabla(P + \frac{1}{2}U^2) + \mathbf{U} \times \mathbf{\Omega} + \nabla \cdot \mathbf{T}. \tag{A2}$$

For a steadily rotating CV, (A2) becomes

$$\frac{\partial \mathbf{U}'}{\partial t} = -\nabla(P + \frac{1}{2}U'^2) + \mathbf{U}' \times \mathbf{\Omega}' - 2\mathbf{\Lambda} \times \mathbf{U}' - \mathbf{\Lambda} \times (\mathbf{\Lambda} \times \mathbf{x}) \tag{A3}$$

as given, for example, by Equation (3.2.10) of Batchelor (1967). Clearly, the Coriolis term is the penultimate one and the last is the centrifugal. These have now to be included in (3). The right side of Equation (A2) appears explicitly in the second term of Noca's (3.49) as

$$\oint_S \mathbf{x} \times \left( \mathbf{n} \times \left[ \nabla(P + \frac{1}{2}U^2) + \mathbf{U} \times \mathbf{\Omega} + \nabla \cdot \mathbf{T} \right] \right) dS \tag{A4}$$

which means that, in addition to swapping $\mathbf{U}$ for $\mathbf{U}'$ and $\mathbf{\Omega}$ and $\mathbf{\Omega}'$ in (3), use of a noninertial CV requires the extra terms

$$-\oint_S \mathbf{x} \times \left( \mathbf{n} \times \left[ 2\mathbf{\Lambda} \times \mathbf{U}' + \mathbf{\Lambda} \times (\mathbf{\Lambda} \times \mathbf{x}) \right] \right) dS. \tag{A5}$$

The general form of the impulse formulation, Equation (3), in a steadily rotating frame of reference thus becomes

$$\frac{\mathbf{F}}{\rho} = -\frac{1}{N-1}\frac{d}{dt}\int_V \mathbf{x} \times \mathbf{\Omega}' dV + \oint_S \mathbf{n} \cdot \left( \frac{1}{2}U'^2 \mathbf{I} - \mathbf{U}'\mathbf{U}' - \frac{1}{N-1}\mathbf{U}'(\mathbf{x} \times \mathbf{\Omega}') + \frac{1}{N-1}\mathbf{\Omega}'(\mathbf{x} \times \mathbf{U}') - \int_V 2\mathbf{\Lambda} \times \mathbf{U}' dV -$$
$$\int_V \mathbf{\Lambda} \times (\mathbf{\Lambda} \times \mathbf{x}) dV - \frac{1}{N-1}\oint_S \mathbf{x} \times \left( \mathbf{n} \times \left[ 2\mathbf{\Lambda} \times \mathbf{U}' + \mathbf{\Lambda} \times (\mathbf{\Lambda} \times \mathbf{x}) \right] \right) dS + \mathbf{F}_v. \tag{A6}$$

$\mathbf{F}_v$ will be neglected as before, as we expect viscous effects to be no more important for rotating than stationary wakes. It is assumed that the vortical wake appears stationary relative to the rotating blades, such that the time derivative of impulse — the first integral on the right-hand side of (A6) — vanishes. The Coriolis force can only have radial and circumferential components and the centrifugal force is radial. The radial components cannot contribute to an axial (or indeed to a circumferential) force, and thus the volume integrals of the pseudo-forces at the end of the first line and start of the second line of (A6) do not

contribute to $T = \mathbf{e}_z \cdot \mathbf{F}$. The axial component of the centrifugal contribution to the pseudo-force surface integral in the second line of (A6) also vanishes by symmetry, and the remaining Coriolis term can be evaluated using the vector identity $\mathbf{a} \times (\mathbf{b} \times \mathbf{c}) = (\mathbf{a} \cdot \mathbf{c})\mathbf{b} - (\mathbf{a} \cdot \mathbf{b})\mathbf{c}$. With $S_D$ denoting the downwind face of the CV, the net contribution, which lies in the axial direction, is

$$-2 \int_{S_D} \lambda w x \, dS \tag{A7}$$

because the term involving $\lambda^2 x^2$ will be equal and opposite at the upwind and downwind faces of the CV.

Thus the reduced form of Equation (A6) is

$$\frac{T}{\rho} = \mathbf{e}_z \cdot \left[ \frac{1}{2} \oint_S \mathbf{n} U'^2 dS - \oint_S \mathbf{n} \cdot \mathbf{U}' \mathbf{U}' dS - \frac{1}{2} \oint_S \mathbf{n} \cdot \mathbf{U}'(\mathbf{x} \times \mathbf{\Omega}') dS + \frac{1}{2} \oint_S \mathbf{n} \cdot \mathbf{\Omega}'(\mathbf{x} \times \mathbf{U}') dS - 2 \int_{S_D} \mathbf{e_z} \lambda w x \, dS \right], \tag{A8}$$

Equation (A8) is now applied to the CVs shown in figure 1, but now rotating at the same $\lambda$ as the rotor. The MD term is the conventional one, derived in any textbook on wind turbine aerodynamics:

$$\int_S u(1-u) dS = \int_{S_D} a(1-a) dS \tag{A9}$$

where $S_D$ denotes the downwind face of the CV. The axial velocity contributes to the second (kinetic energy) term an amount

$$-\frac{1}{2} \int_{S_D} (1 - u^2) dS. \tag{A10}$$

Equations (A9) and (A10) can be combined to give

$$-\frac{1}{2} \int_{S_D} a^2 dS. \tag{A11}$$

The kinetic energy has a contribution from the radial velocity:

$$\frac{1}{2} \int_{S_D} v^2 dS \tag{A12}$$

and the circumferential velocity:

$$\frac{1}{2} \oint_S u_\theta'^2 dS = 2 \int_{S_D} (w^2 + \lambda w x) dS. \tag{A13}$$

It will be assumed that $w = 0$ outside the wake. The radial velocity exiting the horizontal face of the CV at $R_{CV}$ does not contribute to the kinetic energy because $v(R_{CV}) \sim 1/R_{CV}$ so the integral of $v^2$ over the cylindrical face can be made arbitrarily small by making $R_{CV}$ sufficiently large.

The vortex terms are better expressed in the inertial frame, $\mathbf{\Omega} = \mathbf{\Omega}' + 2\mathbf{\Lambda}$, because non-zero values of $\mathbf{\Omega}$ indicate fluid elements that have experienced viscous shear, whereas the apparent vorticity $\mathbf{\Omega}'$ has no such physical interpretation. The vortex terms are equivalently expressed as

$$\left[ -\frac{1}{2} \oint_S \mathbf{n} \cdot \mathbf{U}'(\mathbf{x} \times \mathbf{\Omega}) dS + \frac{1}{2} \oint_S \mathbf{n} \cdot \mathbf{\Omega}(\mathbf{x} \times \mathbf{U}') dS \right] - \oint_S \mathbf{n} \cdot \mathbf{\Lambda}(\mathbf{x} \times \mathbf{U}') dS. \tag{A14}$$

For the vortical wake to appear stationary in the rotating frame, $\mathbf{\Omega}$ must be parallel to $\mathbf{U}'$ everywhere that vorticity is non-zero — that is, vortex lines and streamlines are aligned in the rotating frame — by which $\mathbf{n} \cdot \mathbf{\Omega}(\mathbf{x} \times \mathbf{U}') = \mathbf{n} \cdot \mathbf{U}'(\mathbf{x} \times \mathbf{\Omega})$ and the two integrals in square brackets identically cancel, leaving only

$$-\oint_S \mathbf{n} \cdot \mathbf{\Lambda}(\mathbf{x} \times \mathbf{U}') dS = \int_{S_D} 2\lambda w x \, dS \tag{A15}$$

for $T$. Combining these results leads to

$$\frac{T}{\rho} = \frac{1}{2} \int_{S_D} v^2 dS - \frac{1}{2} \int_{S_D} a^2 dS + 2 \int_{S_D} \left( w^2 + \lambda w x \right) dS + 2 \int_{S_D} \lambda w x \, dS - 2 \int_{S_D} \lambda w x \, dS \tag{A16}$$

Curiously, three integrals with the integrand $2\lambda w x$ have arisen from three different origins: the kinetic energy term and the vortex terms each yield the same integral in the positive sense, and the Coriolis terms lends an equal and opposite contribution. The choice of which two to cancel is arbitrary, but either choice leads to the final equation:

$$\frac{T}{\rho} = \frac{1}{2} \int_{S_D} \left( v^2 - a^2 \right) x \, dx + 2 \int_{S_D} \left( w^2 + \lambda w x \right) dS. \tag{A17}$$

which is identical to (19).

## Appendix B: Derivation of thrust from the unsteady Bernoulli Equation

Assume that, at all times, the fluid elements along a streamline passing through the rotor remain irrotational. The trailing vorticity in the wake is assumed to be infinitely thin, which is equivalent to assuming there is no radial vorticity behind the rotor. The unsteady Bernoulli equation along this streamline is

$$\frac{\partial \phi}{\partial t} + \frac{1}{2}(u^2 + v^2 + u_\theta^2) + p = C \tag{B1}$$

where $C$ is a constant and $\phi$ is the scalar potential. If the wake rotates rigidly with the blades, the unsteady term can be expressed in terms of the circumferential velocity and the rotation rate, $\lambda$:

$$\frac{\partial \phi}{\partial t} = \frac{\partial \theta}{\partial t} \left( \frac{\partial \phi}{\partial \theta} \right) = -\lambda x u_\theta \tag{B2}$$

where the negative sign arises because $\theta$ is evaluated in the moving frame, i.e. $\partial\theta/\partial t = -\lambda$. We now apply Equation (B1) at locations just up- and downstream of the rotor to solve for the pressure difference across the rotor, $\Delta p$. Assuming radial and axial velocities to be continuous across the disk, only the circumferential velocity contributes to $\Delta p$:

$$\Delta p = \frac{1}{2}u_\theta^2 - \lambda x u_\theta. \tag{B3}$$

Substituting $u_\theta = -2w$, and integrating $\Delta p$ over the rotor, we recover Equation (25):

$$\frac{T}{\rho} = \int_{S_D} \Delta p dS = 2\int_{S_D} \left(w^2 + \lambda w x\right) dS. \tag{B4}$$

Since the Bernoulli equation along any one streamline is independent of neighbouring streamlines, the blade-element version of this equation should also hold, i.e.

$$\frac{1}{\rho}\frac{dT}{dx} = 2\int_0^{2\pi} \left(w^2 + \lambda w x\right) x d\theta, \tag{B5}$$

which is Equation (28) in the body of the paper.

**Appendix C: An alternative derivation of the blade element thrust equation**

We present here an alternative derivation of the blade element thrust Equation that highlights another aspect of the role of the pressure on the expanding streamsurfaces. The contributions to the impulse equation for elemental thrust, $dT$, are area integrals over the upwind and downwind faces of the CV $V_2$ used in Section 5 and shown in figure 2. The upwind integrals can easily be rewritten in terms of the downwind ones using continuity. In addition to the blade element versions of the two $w-$containing terms in (19), there are contributions to the integrand from:

- The kinetic energy entering the inlet face of the CV with the value $-\frac{1}{2}(1-a)$,

- The kinetic energy leaving the CV due to $a$ and $v$: $\frac{1}{2}\left((1-a)^2 + v^2\right)$,

- The usual MD term for the annular streamtube, given by $(1-a) - (1-a)^2 = a(1-a)$, and

- The component of kinetic energy normal to the underside and upperside of the annular streamtube.

Only the last is not straightforward. Consider a new CV bounded by the streamsurface passing through the rotor at radius $x_{BE}$ and extending to $R_{CV}$ as before. This CV is outside $V_1$ in in figure 2. There is no thrust and its impulse equation reduces to

$$\int_0^{2\pi}\int_{x_{BE}}^{\infty} \left(v^2 - a^2\right) x dx d\theta = -\int_0^{2\pi}\int_0^{x_{BE}} \left(v^2 - a^2\right) x dx d\theta = \int_0^{2\pi}\int_{x_{0,BE}}^{x_{BE}} \left(1 - (1-a)^2 - v^2\right) x \frac{dx}{dz} dz d\theta \tag{C1}$$

where $x_{0,BE}$ is the radius in the undisturbed upwind flow of the streamsurface passing through $x_{BE}$. The last integrand is evaluated along the streamsurface and must be positive, as must its integral which also gives the non-zero axial component of

the pressure acting on the streamsurface. Therefore, the area integral of $v^2 - a^2$ is *not* zero over the wake; $S_D$ must extend to $R_{CV} \gg 1$ for (24) to hold. Applying (C1) to the upper and lower surfaces of the streamtube, and combining the result with the other three contributions to $dT$, all terms in $a$ and $v$ cancel to recover Equation (28).

*Acknowledgements.* DW's contribution to this work is part of a research project on wind turbine aerodynamics funded by the NSERC Discovery Program. EL acknowledges receipt of an NSERC Post-Doctoral Scholarship. The authors thank the two anonymous referees and Dr Gijs van Kuik for their valuable comments.

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
