# Peer review of "An impulse-based derivation of the Kutta-Joukowsky equation for wind turbine thrust"

_Wind Energy Science, 2019_

## Referee Comment (RC1) · Anonymous Referee #1 · 9 Feb 2020

The paper contains a comprehensive analysis employing the so-called 'impulse equation' to determine an equation for the thrust acting on a wind turbine. The background is a theory developed partly in a PhD dissertation of F. Noca (Caltec, 1997). The main idea is to get rid of using explicitly the pressure when determining the thrust in the axial momentum equation.

The paper inscribes itself in a long line of works dealing with the attempt of formulating a consistent derivation of the differential form of the general momentum equations for rotors. In fact, already Glauert, in his derivation of the general blade-element momentum theory, questioned the use of the standard axial momentum equation, and in 1972 Goorjian proved that the standard form of the axial momentum equation is not valid because it leads to a contradiction when combined with the other equations of the momentum theory. In the abstract of the present paper it stated that 'The new formulation gives a very simple, exact equation for blade element thrust which is the major contribution of this study. By removing the pressure partly through the kinetic energy contribution of the radial velocity, the new equation circumvents the long-standing concern over the role the pressure forces acting on the expanding annular streamtube intersecting each blade element.' However, I am not at all convinced that paper has solved the longstanding concern and that it adds anything new to the already known knowledge on the subject. In the following I will explain.

First, all equations seem to be consistent and I did not find any errors in their derivations. However, it also seems unnecessary complicated to start with a (very) general form of the impulse equation to end up with an expression that has been known for more than a century. Indeed, eq. (38), which, as stated in the conclusion, is being the most significant result of the present analysis, has been known and used by both Joukowsky in his pioneering work on propellers already in the start of the last century and by Glauert in his formulation of the blade–element momentum theory. However, they derived the expression simply by applying the Bernoulli equation in a rotating frame of reference along a streamline going through the rotor plane. This equation is fully consistent with energy conservation and is one of the main equations in the formulation of the Joukowsky propeller theory. In the work of Glauert, the equation corresponding to eq. (38) is written as

$$\frac{\Delta T}{\Delta A} = \Delta p = 2\rho w \left( x\Omega + w \right). \tag{1}$$

Independent of the way of derivation, this equation is (of course) both correct and consistent. However, the equation does not prove that there is no impact of the pressure on the lateral boundaries of the control volume and that the axial momentum theory can be used in its simple form. In its general differential form, the axial momentum equation reads

$$\Delta T = 2\rho a \left( 1-a \right) \Delta A + \oint_{cs} p dA, \tag{2}$$

The problem stated by Glauert (and Goorjian) is if the pressure contribution can be ignored, i.e. if axial momentum theory can be written as,

$$\Delta T = 2\rho a \left( 1-a \right) \Delta A, \tag{3}$$

that is ignoring the impact of the contribution of the pressure forces acting on the expanding streamtubes in the derivation. And here Goorjian was the first to prove that this simple form results in an inconsistency. So the problem is not solved, as the present work only shows that eq. (1) can be derived in different ways, but not that the pressure can be neglected in eq. (2).

My recommendation is that the derivations are maintained in the paper, as they constitute an alternative way of deriving eq. (1), but that the narrative and conclusion that this work solves the 'old' issue of the lack of an exact equation for blade element thrust is taken out of the paper and reformulated. The main problem is not how to relate the azimuthal velocity to the thrust, but how to include the axial velocity, and this problem is not solved in the paper.

---

## Referee Comment (RC2) · Anonymous Referee #2 · 8 Apr 2020

Review of the manuscript "Derivation of an Impulse Equation for Wind Turbine Thrust" by Eric J. Limacher and David H. Wood

The manuscript aims to present a derivation of the thrust of a wind turbine. To do so, the authors use the momentum equation expressed in the "impulse" formulation as presented by Noca et al {Noca, F., Shiels, D., Jeon, D.: Measuring instantaneous fluid dynamic forces on bodies, using only velocity fields and their derivatives. Journal of Fluids and Structures, 11(3), 345-350, 1997.}. The work by Noca et al is relevant to enable determining the loads around a body/force field exclusively by integrating the velocity field (and the derived vorticity field) over a control surface. This is relevant for experimental work when pressure measurements are not available. The authors aim to present a formulation for determining the thrust form the velocity field nearby the

turbine. Such a formulation would be useful. However, the work presented lacks in several aspects: 1- Clarity of the manuscript. a. The paper is mostly clearly written in terms of the use of language. b. The structure of the derivation is not always clear, and terms of the equations are presented or referenced without clear identification. I found it very difficult to follow the derivations, namely because there are no supporting figures and the interpretation of the control surfaces considered is very difficult. 2- The work is not novel in the final derivation presented, as this has been achieved previously by Glauert (I am aligned with the other reviewer in this point) and is clearly presented in {Glauert H. (1935) Airplane Propellers. In: Aerodynamic Theory. Springer, Berlin, Heidelberg} . The derivation by Glauert presents the differential for along the streamline of the variation of the total pressure, which is what the authors now present in an integral approach. 3- Incompleteness of the derivation. In Glauert's work, the jump of stagnation enthalpy is equaled to the jump in pressure and equaled to the thrust. However, the derivation implicitly assumes azimuthal velocities (by the rotational velocity downwind of the actuator), but ignores the effect of vorticity. The authors also neglect the effect of vorticity across the control surface, and that is why they reach the same result. The statement in line 145 " For the vortical wake to appear stationary in the rotating frame, âĎȩ must be parallel to U0 everywhere that vorticity is non-zero— that is, vortex lines and streamlines are aligned in the rotating frame — by which n ÂůâĎȩ(x × U0 ) = n Âů U0 (x ×âĎȩ) and the two integrals in square brackets identically cancel, . . ." implies a rigid wake. Also, even in the case of a rigid wake the statement is not correct, as vortex direction and streamlines do not need to be aligned. It is a convenient simplification, but then we come back to the solution by Glauert. Although the attempt is very interesting and could be useful, the work is not yet complete.
* * *

---

## Author Comment (AC1) · 6 May 2020

We thank both reviewers for their insightful and well-articulated comments and for the time spent checking the derivations of the equations. We have revised the manuscript in response to their important points, and to refocus the work. Changes to the document are highlighted in blue text for ease of reference, and the most important changes are discussed below. Many of the comments were made by both reviewers so we have not separated our responses.

The most important criticism from both reviewers appears to be that the final blade-element thrust equation that we derived:

$$\frac{1}{\rho}\frac{dT}{dx} = 2 \int_0^{2\pi} (w^2 + \lambda wx)x \, d\theta \tag{1}$$

is not novel. This was an oversight on our part, and we have added a reference to Glauert's (1935) work, as well as to Sørensen's (2016) book. We have also modified the title of the paper to reflect this recognition and have added three figures to help clarify the presentation. With that point clarified, the remaining contributions of the work deserve additional emphasis.

Our work addresses three key assumptions of blade-element theory through the lenses of impulse theory and helical vortex theory: (i) the assumption that radial velocity is negligible at the rotor, or at least that it has little effect on blade forces, (ii) that the Kutta-Joukowsky (KJ) equation is applicable to individual blade elements, and (iii) that pressure on the lateral control surfaces does not contribute to blade-element thrust (commonly referred to as the assumption of blade-element independence). These points are now listed explicitly in the first paragraph of the introduction.

The second reviewer has noted that the streamlines and vortex lines need not be aligned in the rotating frame, even for a rigid wake. This is true when diffusion is present, when the circulation of fluid elements in the wake may change over time. However, we have treated the wake as a collection of trailing helicoidal sheets, and it is assumed that Kelvin's theorem holds for individual fluid elements. As such, the vortex lines and streamlines *must* coincide, or else the structure of the wake would appear to deform in the rotating frame (i.e. the wake would not be rigid). A note to this effect is given after equation (12). The insight regarding radial velocity is unique to the present derivation. We have deduced that the radial velocity must be of similar magnitude to the axial induction factor over at least part of the rotor plane, and that this most likely happens in the vicinity of the tip. This agrees with observations by Madsen et al. (2010) and Sørensen (2016, see Figure 3.2). For loaded rotors, the assumption of negligible radial velocity in the power producing region near the blade tips is unjustified, and further work to account for its effect on blade forces is warranted. The discussion of radial velocity is now in section 4.1, as the discussion has been reorganized into subsections.

In response to the first referee comments we have given an expanded discussion on the relationship between the KJ equation (1) and the conventional form of the momentum equation in section 4.1 This also addresses the issue of blade-element independence. Equation (1) above applies to individual blade elements irrespective of the loading on the rest of the blade. Specifically, we have proved in the main text and what is now Appendix C, that the pressure acting on the expanding flow through the blades does not contribute to the axial momentum balance. Of course, blade-element independence as conventionally understood pertains to axial momentum theory (AMT). By taking equation (1) for thrust and showing the conditions under which it collapses to the classical equation from AMT, we have shown the conditions under which that equation is also independent of loading on neighbouring elements. The

following conditions turn out to be sufficient for blade-element independence: negligible viscous drag on the blades; high number of blades and/or high tip speed ratio; and constant vortex pitch across the wake. This conclusion is not unique to the method of derivation equation (1), but rather follows from it and from symmetry arguments from helical vortex theory.

In the original manuscript, we briefly demonstrated that equation (1) is a generalization of the KJ equations for blade elements. This discussion has been expanded in section 4.2, and a figure has been added to show how Stokes' theorem has been invoked.

Now that we have removed claims of novelty regarding equation (1) above, we recognize that the presented derivation is more complicated than strictly necessary to support our conclusions. One motivation to keep the full derivation is for future work. In particular, we are interested in modelling unsteady flows, and for this purpose the unsteady impulse term may prove useful. The shorter of the two derivations of equation (1) has been retained in the body of the work (using a stationary frame of reference), and the other (in the rotating frame) has been moved to Appendix A. We have also added a short analysis of the relationship between thrust and torque, showing that the usual assumption that extracted power equals the thrust multiplied by the velocity through the rotor, is incorrect at low tip speed ratio (TSR) but becomes approximately correct at high TSR (see section 4.3 in the revised manuscript).

Appendix B has been added to show the derivation of equation (1) above using the unsteady Bernoulli equation, upon which previous derivations are based. Its inclusion allows the reader to see that the same assumptions are required to get to (1) whether one uses the Bernoulli equation or impulse theory. In our impulse-based derivation, however, Equation (11) in the revised manuscript is a more general expression which avoids the thin-wake assumption, but this cannot be avoided using the Bernoulli equation. We have presented this as an advantage of the present derivation in the text below Equation (28) in the revised manuscript. Appendix C is the same as the appendix in the original manuscript, with only a couple minor wording changes.

Figures have been added to make the analysis easier to follow, depicting the volumes and surfaces under consideration. Also, in response to a request for clarification from another colleague, we have added a brief introduction to the concept of impulse in the second paragraph of the introduction. This small change should make the paper more accessible to a wider range of wind energy researchers and practitioners.

We hope that our revisions are to the satisfaction of the reviewers, and we look forward to any further comments they have.

---

## Referee Report (RR1)

The authors have done a great job in modifying and correcting the article. However, there are still some (minor) issues that I would like them to consider.

- The paper is at places unnecessary complicated to read and comprehend. I suggest that the authors take a critical look and add some sentences to guide the reader more easily through the paper. An example of this is eqs. (5 ) – (7) on page 5, where eq. (5) is a relation and (6) and (7) are terms, and then it states that 'Equations (5) and (6) can be combined to give..(7)'. This is confusing, how can a relationship (5) and a term (6) be combined to give a term (7)?

- As far as I can see, eq. (12) is only valid along a horizontal stream surface. But that implies that there is no expansion, and then the whole idea breaks down about formulating an alternative version of the momentum theory, as without expansion there are no influence of the pressure on the lateral part of the stream surfaces, and therefore the axial momentum equation can be use without any problems.

- To be honest, then I have a problem in accepting eq. (24). Why should the integral of the radial velocity squared by equal to the integral of the axial induction squared at any arbitrary cross section upstream the rotor? I understand that this is a direct consequence of eq. (19). But, again, as far as I can see, the elimination of the vorticity is based on eq. (12) that assumes no expansion. With no expansion eq. (24) is clearly satisfied as here $v=a=0$. However, since eq. (28) is correct, as it can very easily be derived from the unsteady Bernoulli equation, I suggest that the point is that the right hand side of eq. (24) is outbalanced by the neglected vorticity terms. This, I think is still interesting, and deserves to be reported. In the opposite case, it would be an easy task numerically to show if eq. (24) is valid or not. Furthermore, the argument that eq. (25) is correct if a and v is continuous is somewhat vague, as it foremost demands that eq. (24) is correct at any cross section upstream of the rotor.

- I am not sure that I understand the content of Appendix C. Could you try to explain what control volumes you are using, what is $x_{o,BE}$ and what is the purpose of the appendix?

---

## Referee Report (RR2)

Review of Eric J. Limacher and David H. Wood

An impulse-based derivation of the Kutta-Joukowsky equation for wind turbine thrust, Revised version

**General remarks:**

The authors aim to develop a "Kutta-Joukowsky-equation", Eq. (28), by avoiding use of pressure. As the reviewer's first-round remarks, there are already derivations (or discussions) from Glauert (1935) Soerensen (2016) and van Kuik (2018). Nevertheless, this does not mean that a new derivation is meaningless, but it then should be "easier" to understand or lead to further progress.

Unfortunately, I do not see if this goal has been achieved.

**Specific remarks:**

A complete list of used symbols before the actual text would be desirable.

Figure 1: Please add a coordinate-system

Eqs. (2) and (3): Please add statements about the regularity of the velocity and vorticity fields you imply, i. e. to which class $C^1, C^2 \ldots C^\infty$ they should belong, to make all integrals well-behaved. To put it more in terms of physics: What assumptions of vortex lines, sheets or so on are made implicitly?

Eq. (12) Is this simply a definition of p or a non-trivial statement?

 (An additional sketch would be helpful)

Line 141: "the other dynamic conservation equation" Do you mean: "The other dynamical equation based on conservation of angular momentum"?

Section 3 (lines 141 to 165): think about skipping it because you state " . . .  has not enriched the analysis"

Eq. (24) Line 171: I'm afraid that from this INTEGRAL almost nothing can be concluded for local behaviour unless you make severe assumptions on v(x) and a(x).

Line 174: ". . . likely consequence v ≈ a at the edge of the wake". As in Fig. 15/16 of Madsen et al. (2010) only results from numerical investigations are discussed, an analytical approach should give saver estimates about the flow esp. at the edge (x=1) of the actuator disk.

Line 220:  How are Eq. (30) and Eq. (12) connected ?

Line 233: " . . .  a constant-p wake defines an optimal rotor, (originally from Betz (1919) . . . ).

Unfortunately, Betz, in his paper only investigated lightly-loaded propellers and not heavily-loaded wind turbines, as also remarked in Soerensen (2016).

Line 268/269:  "We are completing .." think about skipping this sentence

Line 217: How can x be large if 0 < x < 1 ?

Line 301 ff: " . . . approximately correct at high tip speed ration . . . "

Can this be made more explicit? Like $O(\lambda^{-n})$ ?

Line 302: " . . .  cannot describe the runaway (raw) state …" As far as I know , this stat (cP = 0, TSR_raw < ≈ ) is excluded from this model at all, because cP increases (strongly) monotonously.

Line 303: "Thus the trailing . . . assumed in lifting line theory for wings." Isn't this statement trivial and superfluous, as pure translational  rotational motion have nothing in common ?

---

## Referee Report (RR3)

Review of WES-2019-93 V3

An impulse-based derivation of the Kutta-Joukowsky equation for wind turbine thrust

by

Eric J. Limacher and David H. Wood

**General comments:**

The manuscript (now in its 3$^{rd}$ version) has been reformulated and extended considerably.

I agree on publishing it if my recommendations from below are included.

**Specific Remarks**

Page 1, line 18:

You mentioned "all texts on wind turbine aerodynamics". This is not completely true. To be update, please add:

 S. Schmitz, Aerodynamics of Wind Turbines, Wiley (2018)

A.P. Schaffarczyk, Introduction to Wind Turbine Aerodynamics, 2$^{nd}$ Ed. SpringerNature (2020)

Page 8, lines 165 to 187.

Pseudo-equations using "≈" should not be present in a scientific paper. Again: Eq (24) is an integral over two functions - to draw any conclusion about the local behaviour demands mathematical assumption in which regularity class the functions $v^2$ (r) and $a^2$ (r) are embedded. Any decisive conclusion can only be drawn from the differential equation.  See G. Gallavotti, Foundations of Fluid Dynamics, section 2.4.

In particular a possible (edge-)singularity at x = 1 may spoil the argument.

Page 2 Eqs. (1) to (2):

If I insert $u_\theta = -2 w$ into Eq. (1) I feel that a factor of 2 is missing in the $w^2$ term of Eq. (2).

Please correct this typo, if it is one.

Page 12, line 261

I do not understand why $\partial \phi / \partial \theta$ equals $x u_{\theta}$. Please explain.

---

## Author Response (AR2)

*We would like to thank the reviewers for their time and their thoughtful comments. Our responses below are in blue text. Changes to the manuscript are also in blue. In response to the second reviewer, we have removed our discussions regarding wake pitch and blade-element independence. This has allowed us to focus more on the blade-element thrust derivation, and the new insights regarding radial velocity. Because the reviewers made overlapping comments, we have not attempted to separate our responses in terms of text colour in the manuscript.*

Reviewer 1:

The authors have done a great job in modifying and correcting the article. However, there are still some (minor) issues that I would like them to consider.

- The paper is at places unnecessary complicated to read and comprehend. I suggest that the authors take a critical look and add some sentences to guide the reader more easily through the paper. An example of this is eqs. (5 ) – (7) on page 5, where eq. (5) is a relation and (6) and (7) are terms, and then it states that 'Equations (5) and (6) can be combined to give..(7)'. This is confusing, how can a relationship (5) and a term (6) be combined to give a term (7)?

  *Thank you for this comment. We have attempted to make the derivation more fluid (pun not intended) throughout. We also believe that removal of content of secondary importance, as noted above, has made our key arguments easier to follow.*

- As far as I can see, eq. (12) is only valid along a horizontal stream surface. But that implies that there is no expansion, and then the whole idea breaks down about formulating an alternative version of the momentum theory, as without expansion there are no influence of the pressure on the lateral part of the stream surfaces, and therefore the axial momentum equation can be use without any problems.

  *We have clarified that our analysis is fully three-dimensional. The simplification of the vortex terms in the near wake is still valid when expansion occurs. Figure 2 has been added to illustrate the coincidence of streamlines and vortex lines in three dimensions. A two-dimensional projection of these vectors leads to Equation (15) in the revised manuscript.*

- To be honest, then I have a problem in accepting eq. (24). Why should the integral of the radial velocity squared by equal to the integral of the axial induction squared at any arbitrary cross section upstream the rotor? I understand that this is a direct consequence of eq. (19). But, again, as far as I can see, the elimination of the vorticity is based on eq. (12) that assumes no expansion. With no expansion eq. (24) is clearly satisfied as here v=a=0. However, since eq. (28) is correct, as it can very easily be derived from the unsteady Bernoulli equation, I suggest that the point is that the right hand side of eq. (24) is outbalanced by the neglected vorticity terms. This, I think is still interesting, and deserves to be reported. In the opposite case, it would be an easy task numerically to show if eq. (24) is valid or not. Furthermore, the argument that eq. (25) is correct if a and v is continuous is somewhat vague, as it foremost demands that eq. (24) is correct at any cross section upstream of the rotor.

  *To derive $\int_{S_U}(v^2 - a^2)dS = 0$ on any plane upstream of the rotor, it is not necessary to "eliminate" the vorticity; the flow upstream is naturally irrotational. The assumption needed to make the integral zero at or behind the rotor plane is that $v$ and $a$ are continuous across the rotor plane. This is enforced to be the case in actuator disk simulations, and we have cited four studies that show our prediction $v \approx a$ agrees with numerical results and one that directly supports the integral above (Equation (24) of the manuscript).*

*In reviewing our discussion of radial velocity, we noticed that the conclusion $\int_{S_U}(v^2 - a^2)dS = 0$ implicitly assumes the flow to be circumferentially uniform. This fact is now highlighted at the beginning of section 4.1.*

- I am not sure that I understand the content of Appendix C. Could you try to explain what control volumes you are using, what is xo,BE and what is the purpose of the appendix?

*We have removed Appendix C.*

Reviewer 2

**General remarks:**

The authors aim to develop a "Kutta-Joukowsky-equation", Eq. (28), by avoiding use of pressure. As the reviewer's first-round remarks, there are already derivations (or discussions) from Glauert (1935) Soerensen (2016) and van Kuik (2018). Nevertheless, this does not mean that a new derivation is meaningless, but it then should be "easier" to understand or lead to further progress.

Unfortunately, I do not see if this goal has been achieved.

*We would like to thank the reviewer for the thoughtful comments. We have sought to clarify our contribution further, improve the organization of our arguments, and omit material of secondary importance. In particular, we emphasize that our derivation of the Kutta-Joukowsky (KJ) equation for blade-element thrust is more general than the classical derivation. Ours permits the wake to be rotational, which the classical derivation based on the Bernoulli equation does not allow. Our derivation also lends novel insight into the magnitude of the radial velocity as documented in the reponse to the first reviewer.*

*Edits to the manuscript are in blue for ease of reference. We have responded to the specific comments in blue text below.*

**Specific remarks:**

A complete list of used symbols before the actual text would be desirable.

*Unfortunately, this does not appear to accord with the journal style. We have endeavoured to clearly define each variable in the text at its first introduction.*

Figure 1: Please add a coordinate-system

*We have shown the coordinate system on the left sketch of figure 1 and have introduced the coordinate variables in the text.*

Eqs. (2) and (3): Please add statements about the regularity of the velocity and vorticity fields you imply, i. e. to which class $C_1, C_2 \ldots C_\infty$ they should belong, to make all integrals well-behaved. To put it more in terms of physics: What assumptions of vortex lines, sheets or so on are made implicitly?

*These equations certainly apply to $C_\infty$ continuous velocity fields, but are often applied to discontinuous velocity fields (flows featuring vortex sheets or discontinuous changes in total head). However, we're not aware of a*

*rigorous proof of their limitations. Your comment raises an important point. Up until Equation (14) in the revised manuscript, we have made no assumptions about discontinuities; the flow is treated completely as $C_\infty$-continuous. At this point in the previous manuscript, we suggested that by treating the wake as a series of thin vortex sheets, we could simplify the vortex terms. In fact, the same simplification can be made without this assumption. We have now refined our argument, assuming only that diffusion on the downstream plane is negligible, so that Kelvin's circulation theorem applies to individual fluid elements. This makes streamlines and vortex lines coincident in the near wake without assuming the wake to be infinitely thin. We have added this line of reasoning to the text after Equation (14). This improvement sets the derivation apart from the classical derivation based on the Bernoulli equation, and we have emphasized this distinction.*

Eq. (12) Is this simply a definition of p or a non-trivial statement?
(An additional sketch would be helpful)

*We have removed all discussion of pitch, and so p has been removed from this equation. We have also added a sketch to illustrate the similarity of velocity and vorticity triangles in the near wake.*

Line 141: "the other dynamic conservation equation" Do you mean: "The other dynamical equation based on conservation of angular momentum"?

*We have removed all discussion of angular momentum and torque to focus more clearly on the thrust.*

Section 3 (lines 141 to 165): think about skipping it because you state " . . . has not enriched the analysis"

*Thank you for this suggestion. We have removed this discussion.*

Eq. (24) Line 171: I'm afraid that from this INTEGRAL almost nothing can be concluded for local behaviour unless you make severe assumptions on v(x) and a(x).

*We agree that we cannot make definitive local statements about $a(x)$ and $v(x)$. However, we have clarified our arguments after Eq. (24) for why $v = a$ is likely to occur in the vicinity of the rotor tip. Firstly, $v = 0$ at the rotor axis, and so $v < a$ over the inner rotor, but we also expect $v > a$ outside the wake. Thus, $v = a$ must occur somewhere, and seems likely that this would happen where $a$ is rapidly decreasing at the edge of the wake. We have referred to four numerical studies that support these results and one that specifically verifies $\int_{S_D}(v^2 - a^2)\, dS = 0$ at the rotor plane.*

*While reviewing our arguments here, we also noted that the result $\int_{S_D}(v^2 - a^2)\, dS = 0$ carries the implicit assumption of circumferential uniformity. We have made note of this fact at the beginning of section 4.1.*

Line 174: ". . . likely consequence v ≈ a at the edge of the wake". As in Fig. 15/16 of Madsen et al. (2010) only results from numerical investigations are discussed, an analytical approach should give saver estimates about the flow esp. at the edge (x=1) of the actuator disk.

*Yes, the results of Madsen et al. (2010) are numerical. Whether our analytic arguments or their numerical results are "safer," we have not said. But the results of both approaches are consistent.*

Line 220: How are Eq. (30) and Eq. (12) connected?

*Equation (30) has been removed.*

Line 233: " . . . a constant-p wake defines an optimal rotor, (originally from Betz (1919) . . . ).

Unfortunately, Betz, in his paper only investigated lightly-loaded propellers and not heavily-loaded wind turbines, as also remarked in Soerensen (2016).

*You are correct, and this limitation does restrict the scope of our arguments more than we would prefer. In favour of sharpening our stronger arguments, we have omitted all of the discussion regarding wake pitch.*

Line 268/269: "We are completing .." think about skipping this sentence

*Done.*

Line 217: How can x be large if $0 < x < 1$ ?

*We were referring to locations near the tip, i.e. $x \rightarrow 1$. However, we have removed this discussion altogether.*

Line 301 ff: " . . . approximately correct at high tip speed ration . . . "
Can this be made more explicit? Like $O(\lambda_{-n})$ ?

*This discussion has been removed.*

Line 302: " . . . cannot describe the runaway (raw) state …" As far as I know, this stat ($cP = 0$, $TSR\_raw < \approx$ ) is excluded from this model at all, because cP increases (strongly) monotonously.

*This discussion has been removed.*

Line 303: "Thus the trailing . . . assumed in lifting line theory for wings." Isn't this statement trivial and superfluous, as pure translational rotational motion have nothing in common?

*This statement alludes to future work that we are conducting, and we recognize that this statement does not add much to the central arguments we are making. It has been removed.*

---

## Author Response (AR3)

Review of WES-2019-93 V3
An impulse-based derivation of the Kutta-Joukowsky equation for wind turbine thrust
by
Eric J. Limacher and David H. Wood

**General comments:**
The manuscript (now in its 3rd version) has been reformulated and extended considerably.
I agree on publishing it if my recommendations from below are included.

Thank you, once again, for your useful feedback. We have responded to each of your comments below, and the changes in the revised manuscript are in red text for ease of reference.

**Specific Remarks**
Page 1, line 18:
You mentioned "all texts on wind turbine aerodynamics". This is not completely true. To be update, please add:
S. Schmitz, Aerodynamics of Wind Turbines, Wiley (2018)
A.P. Schaffarczyk, Introduction to Wind Turbine Aerodynamics, 2nd Ed. SpringerNature (2020)

Thank you for suggesting these recent citations. They have been included in the first paragraph of the introduction. We have also changed "all texts" to "many texts."

Page 8, lines 165 to 187.
Pseudo-equations using "≈" should not be present in a scientific paper. Again: Eq (24) is an integral over two functions - to draw any conclusion about the local behaviour demands mathematical assumption in which regularity class the functions $v^2$ (r) and $a^2$ (r) are embedded. Any decisive conclusion can only be drawn from the differential equation. See G. Gallavotti, Foundations of Fluid Dynamics, section 2.4.
In particular a possible (edge-)singularity at x = 1 may spoil the argument.

You are correct to point out that our arguments depend on the assumption that $v^2$ and $a^2$ are at least C0-continuous. We have now made that assumption explicit in the modified paragraph on pg. 8. With that made clear, we can definitively argue that $v^2=a^2$ somewhere on S_U, once we note that v=0 at x=0 by symmetry while a$\geq$0 at x=0. We then point the reader to three studies employing actuator disk simulations that show v=a occurs near the rotor tip for a wide range of thrust coefficients.

We have likewise altered the corresponding claims in the abstract and conclusion. We claim only to show that the magnitudes of v and a are equal somewhere on the plane containing the rotor, which previous studies have shown to occur near the rotor tip over a wide range of thrust coefficients.

Page 2 Eqs. (1) to (2):
If I insert $u_\theta$ = -2 w into Eq. (1) I feel that a factor of 2 is missing in the $w^2$ term of Eq. (2).
Please correct this typo, if it is one.

Yes, this was a typo. Thank you for catching it. It has been corrected, and Equation (2) is now consistent with Equation (25).

Page 12, line 261
I do not understand why $\partial \phi / \partial \theta$ equals $x u_{\theta}$. Please explain.

In the unsteady Bernoulli equation, $\phi$ is the scalar potential, related to the velocity vector according to $\boldsymbol{U} = \boldsymbol{\nabla}\phi$. The azimuthal component of velocity is thus $u_\theta = \frac{1}{x}\frac{\partial \phi}{\partial \theta}$. A note to this effect has been added after Equation (A2).